# Long-Range Wireless System for U-Value Assessment Using a Low-Cost Heat Flux Sensor

**DOI:** 10.3390/s22197259

**Published:** 2022-09-25

**Authors:** Marc Lazaro, Antonio Lazaro, Benito González, Ramon Villarino, David Girbau

**Affiliations:** 1Department of Electronics, Electrics and Automatic Control Engineering, Rovira i Virgili University, 43007 Tarragona, Spain; 2Institute for Applied Microelectronics, Campus Universitario de Tafira, Universidad de Las Palmas de Gran Canaria, 35017 Las Palmas de Gran Canaria, Spain

**Keywords:** heat flux sensor, U-value, LoRa, thermal isolation, 3D printed, low cost

## Abstract

The present study exposes an economical and easy-to-use system to assess the heat transfer in building envelopes by determining the U-value. Nowadays these systems require long wires and a host to collect and process the data. In this work, a multi-point system for simultaneous heat flux measurement has been proposed. The aim is to reduce the long measurement time and the cost of thermal isolation evaluations in large buildings. The system proposed consists of a low-cost 3D-printed heat flux sensor integrated with a LoRa transceiver and two temperature sensors. The heat flux (HF) sensor was compared and calibrated with a commercial HF sensor from the Fluxteq brand.

## 1. Introduction

With the relentless increase in energy consumption in society, the improvement of the efficiency and sustainability of the human lifestyle is increasingly being sought. The residential sector is currently the second largest energy sink, with 26.1% of total energy consumption [1]. Breaking down energy consumption in homes by type of end-use, more than half of the energy is used to heat spaces [2]. Therefore, the reduction of part of the heating consumption of homes through the evaluation and improvement of the thermal insulation of buildings means a major step towards the transition to a sustainable future.

The thermal performance of buildings is assessed, in terms of heat loss, by means of the thermal transmittance (U-value) of the building envelope. U-value is expressed in W/(m${}^{2}$·K). Each country stipulates its own insulation regulations regarding energy conservation and resource consumption according to its climatic conditions. These requirements are an upper bound of the U-value for each element of the building (e.g., walls, floors, ceilings, windows).

Although these regulations apply to new construction and rehabilitation projects, there are many works that were carried out with less strict regulations or, sometimes, without any. Due to uncertainty in the type of some materials used in older buildings, on-site measurement of thermal transmittance is the only way to assess its thermal performance. Currently, there are several methods accepted by the international organization for standardization (ISO) for determining the U-value.

The calculation method, described in the ISO 6946:2017 standard, consists of theoretically approximating the U-value by measuring the width of the layers within the building envelope [3]. This method generally requires destructive procedures, creating holes and cavities. The U-values obtained using this method can be unrealistic due to the inhomogeneities of layers and construction materials.The heat flux method (HFM), described in the ISO 9869-1:2014 standard, consists of measuring the thermal transference by means of a heat flux-meter and a pair of temperature sensors [4]. This method provides a reliable U-value if the wall monitoring and data analysis requirements described in ISO 9869 are accomplished. Some of the most significant requirements are that the heat flux probe cannot be installed near thermal bridges or partially in contact with surfaces and sensors cannot be exposed to direct sunlight radiation or rapidly changing weather conditions. To ensure stable conditions and reliable results, measurements should be performed over a period of at least 72 h. The HFM is still considered a more accurate procedure to assess the thermal transmittance of building envelopes and it is standardized in ISO 9869, ASTM C1046-95, and ASTM C1155-95 [5,6].Infrared thermography, described in the ISO 9869-2:2018 standard, consists of measuring the thermal transmittance of building elements by pointing an infrared (IR) camera at their surface [7,8]. This method makes it possible to monitor large areas by performing a single non-intrusive measurement, and easily detect thermal bridges or non-homogeneous thermal surfaces [9]. Although infrared thermography has aroused great interest during the last decade due to the aforementioned advantages [10], its U-value accuracy highly depends on the device and measurement conditions, the correction of ambient temperature [11] and the equipment to perform the evaluation, which is expensive and requires additional signal processing to process the infrared thermal images. For the latter reasons, infrared-based techniques are commonly used for the qualitative diagnosis of insulated building walls [10]. However, recent advances in passive infrared technology, have made it increasingly accessible, so that new methods for the automatic classification of thermal defects in buildings have been introduced [11,12,13]. In [12], a technique that combines images in the thermal and visible spectra to perform this automatic diagnosis is proposed. Recently, deep learning techniques (mask region-based convolutional neuronal networks) have been applied to thermal images for the automatic inspection of problems related to water and thermal bridges in infrastructures [13].

Apart from the methods mentioned above, some works propose a non-intrusive method similar to HFM based on temperature sensors (TBM). The TBM is a non-standardized method based on Newton’s law of cooling, where the heat flux is approximated through the air temperature and the wall surface temperature. Both the HFM and the TBM require a minimum temperature gradient between the inside and outside temperatures to accurately determine the heat flux. The main difference is that the TBM requires a minimum gradient of 15 °C, and the HFM can be applied from temperature differences greater than 5 °C [14,15,16].

Wireless U-value measurement systems based on commercial Zigbee networks have recently been presented in [15,17]. However, wall attenuation and multipath effects cause large indoor fading. The typical transmit power of a low-power Zigbee node (e.g., DIGI XZBee module) is of the order of 8 dBm, and the receiver sensitivity (for 1% packed error rate PER) is −102 dBm. In a Zigbee network, using a single-hop communication, with a transmission power of 3 dBm to optimize consumption, distances of up to 40 m can be reached indoors [18]. Generally, in large buildings, longer communication ranges are required. The range of the Zigbee network can be increased by implementing multi-hop communication topologies (adding intermediate nodes), which implies an increase in the cost of the system, as well as in the installation time. In addition, interference can occur between Zigbee networks operating in the 2.4 GHz industrial, scientific, and medical radio band (ISM band) and Wi-Fi networks operating in the same or adjacent frequency channels, which usually overlap Zigbee channels due to the wide bandwidth of Wi-Fi communications [19].

The purpose of this paper is to design an economical and reliable U-value wireless measurement system without the range and cost limitations of other commercially available solutions. The proposed system integrates a heat flux (HF) sensor, two temperature sensors, and a processing unit with wireless capabilities. Each of the individual nodes can assess the U-value at the point of installation using the HFM described in ISO 9869-1:2014 standard. In a recent work [20,21], the authors have presented a smart mask based on a double heat flux sensor for the accurate measurement of body temperature. This flux sensor is modified in this paper for U-value measurement.

To improve the communication range available up to now in U-value wireless measurement systems, the use of long-range wide-area networks (LoRaWan or LoRa) transceivers is proposed [22]. Each node collects sensor data and sends it to a gateway. As a result, a multi-point U-value assessment can be performed, drastically reducing measurement time. LoRa operates in the 868 MHz band, thus reducing wave attenuation and avoiding interference from the massively used 2.4 GHz band. Moreover, the high sensitivity of LoRa receivers, which use chirp spread spectrum techniques, allows a communication range of several kilometers in line-of-sight (LoS) environments [23] and works correctly in high path loss environments. The authors have already demonstrated the feasibility of indoor communications using LoRa backscatters for location applications [24,25] and the feasibility to read data from deep implanted devices [26].

The popularity of LoRa devices for Internet-of-Things (IoT) applications makes it possible to find commercial low-cost devices. Therefore, LoRa communication is a good option for this application, not only from a technical point of view but also from an economic point of view. In addition, the cost of commercial heat flux sensors is usually above EUR 100, and the kits to determine the U-value cost above EUR 1000. To overcome these high costs, a 3D-printed HF sensor has been designed, manufactured, and compared to a commercial sensor. The cost of the proposed HF sensor does not exceed EUR 5. The overall cost of each measurement node (including the microcontroller, temperature sensors, and wireless transceivers) is under EUR 40.

This paper is structured as follows. In Section 2, the overall system operation is described. Section 3 presents the design and calibration of the 3D-printed heat flux sensor. Section 4 is devoted to the U-value measurements. Finally, the conclusions of this work are presented in Section 5.

## 2. System Overview

The most reliable system for evaluating the thermal insulation of building envelopes, provided that the measurement conditions described in current standards are met, is by means of the heat flux meter (HFM) method. The system consists of a network of nodes capable of measuring the U-value simultaneously in several points. Each node integrates an HF sensor and two temperature sensors.

Figure 1 describes the general operation of the system. Each node is powered by a battery and does not rely on a computer to collect and transmit data. Any of the nodes with Internet access can act as a gateway, collecting data from all the active nodes. Data are uploaded via message queuing telemetry transport (MQTT) protocol and processed by a host, allowing measurement monitoring in real time. Each node also stores data locally on a security digital (SD) card. Point-to-point communication between nodes is done through LoRa modulation. Data from all sensors are uploaded via Wi-Fi. Typically, the outdoor temperature sensor is wired up to the indoor measurement point, which requires the use of long cables or the installation of the measurement node near windows or doors, where it is common to find thermal bridges. To avoid this problem, wireless communication with the outdoor temperature sensor is proposed. Communication with the outdoor temperature sensor can be done via Wi-Fi or Bluetooth low-energy (BLE), the latter being preferable due to its lower power consumption.

### 2.1. Measurement of the U-Value

The thermal behavior of buildings is measured in terms of heat loss, which is commonly evaluated in the building industry as a U-value, or thermal transmittance (reciprocal of the R-value). The U-value gives a measure of the effectiveness of a material (e.g., a window glass), or an element composed of several materials (e.g., a wall), as a thermal insulator. The lower the U-value, the better the material is as a heat insulator. The U-value of a building component like a wall, roof, or window, measures the amount of energy (heat) that is lost through a square meter (m${}^{2}$) of that material for each degree (K) difference in temperature between the inside and the outside. Under one-dimensional heat flux, the thermal transmittance, or the U-value, is defined as the ratio between the heat flux per unit area, *q* (W/m${}^{2}$) and the temperature difference between each side of the wall, $T_{i}-T_{e}$.
(1)$$U\left(\frac{W}{m^{2}K}\right)=\frac{q}{T_{i}-T_{e}}$$

The heat flux through a building depends on the temperature difference on both sides (inside and outside), and on the conductivity and thickness of the materials used. If the material is a composite (consisting of several material elements), the overall resistance *R* is the sum of the resistances of each element. The U-value can be calculated theoretically as the inverse of the sum of the thermal resistance of the layers that makes up the material:(2)$$U=\frac{1}{R}=\frac{1}{R_{si}+\sum_{i=1}^{n=L}\frac{t_{n}}{k_{n}}+R_{se}}$$
where $t_{n}/k_{n}$ is the thermal resistance of each layer n, $t_{n}$ is the thickness of the material, and $k_{n}$ is its thermal conductivity. *L* is the number of layers. $R_{si}$ and $R_{se}$ are the internal and external thermal surface resistances which typically are 0.13 K/(Wm${}^{2}$) and 0.04 K/(Wm${}^{2}$) [27], respectively.

The U-value is measured according to the ISO 9869:2014 standard [4] using the average method from the setup shown in Figure 1. A heat flux sensor is installed on the wall and the indoor and outdoor air temperatures are monitored by two temperature sensors. The heat flux sensor must be in contact with the surface of the envelope (typically a wall) and it is located inside because the temperature is more stable than outside. Thermal transmittance is obtained by dividing the average heat flux by the average temperature difference (between inside and outside), measured over a continuous period (several days).
(3)$$U=\frac{\sum_{j=1}^{N}q_{j}}{\sum_{j=1}^{N}\left(T_{ij}-T_{ej}\right)}$$
where $q_{j}$ and $T_{ij}$ and $T_{ej}$ are the heat flux samples and the internal and external temperature samples, respectively. *N* is the number of measures made during the test.

The accuracy of the measurements depends on several factors, such as the temperature difference (the higher the temperature, the more accurate the measurement will be), the weather conditions, the system for hooking the heat flux sensor to the test area (a thermal sheet should be used), the measurement point (it should be placed away from thermal bridges), and the duration of the measurement (the probability of reaching a stable state will be greater in case of increasing the duration of the measurements, and consequently the results will be reliable and repeatable). Additionally, convection currents can introduce irregularities in heat flux measurements. Therefore, the installation point of the HF sensor is not trivial, and all the parameters mentioned must be taken into account.

### 2.2. Wireless System Design

Accurate assessment of large areas or buildings requires many measurements at different points, which is time-consuming, especially if these measurements are performed sequentially. Therefore, one of the main contributions of this research is to reduce the cost of the equipment and expand the communication range between nodes to ensure a parallel measurement, thus reducing the evaluation time.

It is very common that in newly built buildings, where energy certificates and thermal insulation evaluation must be carried out, there is no internet connection available. Even if an access point is available, the range of the access point may not be enough to provide coverage to each one of the measurement nodes. In this situation, where only one access point is available or even none at all, it is mandatory to have wireless peer-to-peer (p2p) capabilities in the metering nodes.

Wi-Fi and Bluetooth communication systems work well at ranges of 10 to 30 m, but these distances may not be enough for the evaluation of large buildings, where nodes may be separated by much greater distances. Zigbee networks are an interesting alternative to increase the communication range since they can use mesh topology and multi-hopping communication. Furthermore, the range of the transceivers is higher than that of Wi-Fi and Bluetooth, reaching distances up to 100 m in LoS. Unfortunately, Zigbee networks have some drawbacks as well. The range of the transceivers is abruptly reduced indoors, in many cases forcing one to install several intermediate nodes to ensure the reception of information, which will increase the cost and time of the installation. Therefore, to avoid the aforementioned range limitation, without needing the deployment of a mesh network with intermediate nodes, in this work, communication through LoRa transceivers is proposed.

LoRa is the RF modulation used in LoRaWAN, a low-power wide-area (LPWA) networking protocol designed for the Internet of Things (IoT) [28]. Referring to the open systems interconnection (OSI) model, LoRaWAN is the medium access control sublayer, and LoRa is the physical layer. The first one manages the communication between all the elements of the network to ensure good performance (channel selection, topology, frame structure, security keys, etc.), and the second one defines how the data are modulated and transmitted to the physical medium. Unlike other licensed LPWA systems such as narrowband IoT (NB-IoT), the LoRaWan specification does not define any commercial model or type of deployment. Therefore, it can be used without any license from mobile operators.

The LoRa modulation is based on the chirp spread spectrum (CSS) technique, which has many advantages that makes it ideal for this application. Among all of them, two stand out: the communication range of up to 5 km in urban areas and 15 km in LoS, and its low power consumption is perfect for devices that work with batteries. LoRa receivers have better sensitivity than SigFox, down to −137 dBm. Therefore, they are robust against the attenuation introduced by the walls in buildings [29,30,31]. For the application proposed in this paper, long-range and long-term monitoring is required, in addition to very low data throughput. Therefore, LoRaWAN seems the most suitable solution among all those available.

In order to test the performance of LoRa in a real environment, several transceivers have been deployed on a university campus as, shown in Figure 2. The transmitters constitute the measurement nodes, and the receiver acts as a gateway that collects the data from each node. The LoRa transmitter was configured with an output power of 20 dBm, a spreading factor of 12, and a bandwidth of 125 Hz. The channel frequency used was 868 MHz. Table 1 shows the distance between the transmitters and the receiver, as well as the average received signal strength indicator (RSSI), the average signal-to-noise ratio (SNR), and the percentage of packets lost during the transmission.

The gateway is the node that gathers the data from all the measurement nodes and uploads it, via WiFi, to the server. Therefore, any node that is within range of an internet access point can operate as a gateway, and it is possible to assign this task dynamically to these nodes depending on the strength of the WiFi signal instead of predefining it statically. P2P communication helps to counter interference as there are multiple communication links between nodes.

The message queuing telemetry transport (MQTT) protocol is used for transmitting the data from the gateway to an MQTT broker. The gateway publishes the data corresponding to each node in a topic identified by the identifier of the sensor node. MQTT clients can subscribe to the topic of each sensor node to get the data and store it in a database. The MQTT broker is a server that receives all messages from the MQTT clients and then routes the messages to the appropriate subscribing clients. A free online MQTT broker can be used, but often a private solution is preferred and one’s own MQTT broker is used. In this work, the Mosquitto MQTT broker has been installed on a PC computer. This broker also can run on other low-cost platforms such as a Raspberry Pi.

Communication between each measurement node and its respective outdoor temperature sensor is carried out via Bluetooth low energy (BLE). WiFi could be used for this communication, but Bluetooth is more efficient for low data rate communications, reducing the outdoor sensor’s battery capacity. In short, LoRa modulation is used for the P2P communication between nodes to collect all the data in multi-point measurements, the gateway node uses WiFi to upload the data to the host, and finally, BLE is used in the communication between the metering node and its outdoor temperature sensor. In the tests carried out, an ESP32 DevKitC was used. The current consumption of this board when Bluetooth BLE is enabled is about 100 mA, while the current consumption in deep-sleep mode is approximately 5 mA. There are other options such as the FireBeetle ESP32 board or the WiPy 3.0 board, both based on the ESP32, which can be used with low-drop regulators, and they allow reaching a deep-sleep current consumption of about 150 μA. Another alternative is to use low-power Bluetooth BLE to reduce the consumption of the external module.

Since temperature and heat flux vary slowly over time, the time between measurements can be extended quite a bit to increase battery life, especially in cases where a power outlet is not accessible. Therefore, a duty cycle of about one second per minute is more than enough to track temperature. The rest of the time, the microcontroller is set to deep-sleep mode, therefore the power consumption is drastically reduced to a few μA (about 200 μA on the used TTGO board, of which 50 μA correspond to the quiescent current of the regulator). In long-range transmissions, the SX1276 LoRa transceiver can be configured with a transmission power of 20 dBm and current consumption of 120 mA. However, in the event that the link distance is shorter or in LoS (light-of sight) conditions, the transmission power can be configured up to 13 dBm with a current consumption of 29 mA. The typical current consumption of the board is about 50 mA when the ESP32 microcontroller is in active mode. Table 2 shows the estimation of the battery lifetime for indoor and outdoor wireless systems. The gateway is usually installed at points where power is available, making it less critical. The estimate shows that the system can run on LiPo batteries (commonly available) for the three days required to perform the U-value measurements.

### 2.3. U-Value Meter Prototype

Figure 3 shows a diagram of the U-value metering node. The chosen microcontroller (MCU) is an ESP32 mainly due to two features: the ultra-low-power coprocessor and the hybrid WiFi and Bluetooth chip. The LoRa communication is provided by an SX1276 transceiver, which is controlled via a synchronous serial peripheral interface (SPI). The 3D-printed HF sensor is connected to the MCU via the I2C interface. In order to determine the U-value, in addition to the heat flux, two sensors located on both sides of the wall are needed to measure the ambient temperature. For this reason, an NTC thermistor mounted in a voltage divider topology with a 10 kΩ resistor has been used. The voltage divider output is directly connected to one of the 12-bit analog-to-digital converter (ADC) input channels of the MCU. To correct the ESP32 ADC linearity errors a previously calibrated interpolation table is used to convert the ADC samples to voltage. Since the signal at the output of the divider coincides roughly in the middle of the ADC’s range, these errors are minor and a simple offset correction can be used. A 100 nF capacitor is added at the input of the ADC to filter noise and average the measurement. The outdoor temperature sensor uses the same topology but has its own microcontroller to collect and wirelessly send the temperature measurement to the indoor node. The system includes a memory module to store data locally. MCU saves data from the sensors to the SD card via SPI. Both the indoor node and the outdoor sensor are powered by a battery.

Figure 4 shows a photograph of the prototype. The system components are a TTGO LORA32 board that integrates the ESP32 microcontroller with the SX1276 LoRa transceiver, which operates at the 868 MHz band; a CATALEX microSD memory module v1.0; two radial glass NTC thermistors, more specifically the G10K3976, for the indoor and the outdoor temperature sensors; finally, two MAX30205 temperature sensors that are part of the 3D-printed HF sensor. The overall cost of the sensors is below EUR 40.

The microcontroller acquires the data, saves it locally into an SD card, and finally uploads it via wireless to a server where the measurements can be monitored. Wireless communication may not be strictly necessary if the data are already stored locally. However, due to the long duration of the measurement, with a minimum period of 72 h, the immediate detection of errors can avoid repeating the entire measurement again. Therefore, constant monitoring is required, and this is where wireless communication comes into play.

## 3. Heat Flux Sensor Design

In this section the HF sensor design is detailed. Section 3.1 is devoted to the theoretical basis of the sensor and its principle of operation. Subsequently, Section 3.2 presents the simulations of the sensor. Finally, its calibration process and measurements are presented in Section 3.3.

### 3.1. Operation Principle

The design of the HF sensor is based on the principle of heat transfer by conduction. Energy can be transferred by conduction, convection, or radiation, but when evaluating the thermal insulation behavior of walls, conduction is often the most relevant mode of heat transfer. According to the differential form of Fourier’s law of thermal conduction, the rate of heat transfer is proportional to the product of the local temperature gradient and the thermal conductivity of the material, as can be seen in (Equation 4),
(4)q=−k∇T
where *k* is the thermal conductivity of the material in W/(m·K), and ∇T is the temperature gradient in K/m. The temperature gradient can be expressed in its one-dimensional form dT/dx, where *x* is the heat transfer direction as shown in Figure 5. The heat flux *q* can be written as a function of the thermal resistance $R_{t}h$ and the temperature difference between the two sides:(5)$$q=\frac{\nabla T}{R_{th}A}$$
where *A* is the area of the sensor. Therefore, for a known thermal resistance, the heat flux can be obtained from the measured temperature difference between the two sides, each measured with a temperature sensor.

Figure 5 shows a diagram of the sensor. Because the sensor is made of materials with a known thermal conductivity, the heat flux through the sensor can be determined, since the parameters of Equation (Equation 5) are known. Considering the conservation of energy, or the first law of thermodynamics, if the sensor is placed on a surface, such as a wall, the heat flowing out will be equal to the heat flowing in, thus obtaining the heat flux from the wall.

Recent advances in additive manufacturing technologies and the availability of desktop 3D printers on the market allow designs and prototypes to be manufactured quickly and cheaply [32]. Therefore, 3D printers are widely used in both academia and industry for this purpose. Three-dimensional printing (referred to as additive manufacturing) is a new material processing technology that enables the creation of a physical 3D object from computer-aided (CAD) modeling software. There are several additive manufacturing processes [33,34]. However, the most popular desktop 3D printers for prototyping use fused deposition modeling (FDM), also known as fused filament fabrication (FFF). FDM builds parts layer by layer by selectively depositing melted material in a predetermined path, and uses thermoplastic polymers that come in the form of filaments [32,35]. Thermoplastic materials for FDM include polymers such as PLA (poly-lactic acid), ABS (acrylonitrile butadiene styrene), PETG (polyethylene terephthalate glycol), ASA (acrylonitrile–styrene–acrylester), and PEEK (polyetheretherketone). PLA filament is the most common material used in desktop FDM printers. Printing with PLA is relatively easy and can produce parts with finer details. ABS has superior mechanical properties to PLA, as well as being more durable and lighter, although it is more difficult to be printed. For applications that require higher temperatures, ABS (glass transition temperature of 105 °C) is preferable to PLA (glass transition temperature of 60 °C). PLA can rapidly lose its structural integrity and can begin to warp as it approaches 60 °C. Another advantage of PLA is that, although it is stable under general atmospheric conditions, it is a biodegradable bio-based polymer (made from the fermentation of natural and renewable resources such as corn or sugarcane) while ABS is not biodegradable and more toxic than PLA [36]. PETG is a recyclable polymer that has better shock resistance and has a glass transition temperature (85 °C) higher than PET. PETG is used in food packaging and medical applications and is more durable and more flexible than PLA. Another option similar to ABS is to use ASA, which has a lower glass transition temperature (100 °C) and is characterized by being more resistant to weathering and UV rays than ABS. PEEK is a high-performance engineering plastic with outstanding resistance to harsh chemicals, and excellent mechanical strength. PEEK is suitable for continuous use at temperatures up to 170 °C. In this work, the FDM 3D printer WITBOX2 model from BQ is used for the fabrication of prototypes and cases, which normally allows printing layers with a thickness of 0.1 mm. Table 3 shows the thermal properties of some thermoplastics used in 3D printing. The values in the table have been collected from references in the literature and technical data sheets [37,38,39]. There are no significant differences in thermal conductivity between the different thermoplastic polymers, but there are in hardness and ease of manufacturing, as described above. When measuring the U-value, the flow sensor is installed inside buildings, so that the measurement is not affected by weather conditions or sun exposition. Therefore, the HF sensor is relatively protected and the temperature range to which it must be subjected is lower than outside. Therefore, in this work, as a proof of concept, PLA has been chosen for the realization of the prototypes due to the greater printing facilities compared to other thermoplastic polymers. In addition, it has the lowest thermal conductivity (0.13 W/m·K), and, therefore, better insulation, facilitating greater temperature differences between both sides of the sensor.

The HF sensor consists of a solid PLA cylinder with a diameter of 5 cm, and a width of 1 cm. The temperature at each side of the cylinder is measured using two MAX30205 temperature sensors from Maxim Integrated, which have an accuracy of ±0.3 °C within the range of 15 °C to 45 °C. The sensors are mounted on a Rogers 4003c substrate with a thickness of 32 mils. The bottom layer of copper (Cu) has been kept, ensuring a homogeneous distribution of heat on the sensor surface. Figure 6 shows an image of the prototype.

### 3.2. Sensor Simulation

The HF sensor was simulated by solving the heat equation using a commercial technology computer-aided design (TCAD) tool (Sentaurus Device by Synopsys) [40]. To perform the three-dimensional (3D) numerical simulation, the sensor was placed onto an aluminum sheet with a thickness of 1 mm. The thermal conductivities of materials used in the simulation are shown in Table 4. The heat generated within the HF sensor was considered to be dissipated by convection from the lateral and top side of the sensor and the top side of the aluminium sheet; the convective film coefficient was assumed to be 7.5 W/m${}^{2}$·K [41]. The other boundary conditions were adiabatic ones, except for the isothermal condition at 46 °C at the bottom of the aluminum sheet. The ambient temperature was taken as 25 °C. Thus, Figure 7 and Figure 8 show the resulting temperature distribution in a vertical cut of the sensor and the corresponding temperature increase in depth (along the axis of the sensor), respectively. Then, the thermal resistance $R_{th}$ of the sensor is determined by dividing the temperature difference between points A and B, ΔT = 9.6 °C (see Figure 8), where the temperature sensors are located, and the heat flowing through the sensor, 444 mW, resulting in 21.6 K/W.

### 3.3. Sensor Calibration

To calibrate the 3D-printed sensor, a commercial PHFS-01 heat flux sensor from the brand Fluxteq is used. The PHFS-01 sensor is composed of a series array of Type-T thermocouplers with a sensitivity of 1.37 μV/(W/m${}^{2}$). The measurements for the commercial sensor are performed using the COMPAQ DAQ system from Fluxteq, which integrates a differential amplifier and a 24-bit Delta-Sigma analog-to-digital converter (ADC) to measure the small voltage difference at the output of the T-type thermocoupler array. The sensor is calibrated in the laboratory using a conduction calibration system. To do so, artificial heat flux is applied to both sensors using a Stuart UC150 heater. Both sensors were placed above the heater and in close proximity to each other to ensure that they were both exposed to the same heat fluctuations. The sensors were placed on top of the heater by placing a thermal interface sheet between the two, thus reducing the air gaps and improving the heat transfer. During the measurements, a large plastic box covered the heater and sensors, to keep the ambient temperature as constant as possible. Figure 9 shows a photograph of the equipment used for calibration.

To calibrate the sensor the thermal resistance must be obtained. Even if materials with a known thermal conductivity are used, the micro air gaps generated during the printing process, or the variations in thermal conductivity of PLA depending on the filament manufacturer, can cause the thermal conductivity of the PLA sensor to differ from its theoretical value, which is 0.13 W/m·K. For this reason, it is necessary to experimentally determine the thermal resistance of the sensor to perform its calibration.

To this end, the heater was adjusted to different temperatures, and once the steady state (stable temperature) was reached, the temperature differences between both sides of the 3D-printed sensor and the heat flux were recorded. The heat flux is provided by the commercial sensor from Fluxteq, and the temperature gradient is provided by the two MAX30205 sensors integrated on each side of the 3D-printed sensor. Once recorded, the average temperature gradient and the average heat flux are determined. Thanks to the averaging of the measurements, the error is reduced, which is essential, because the measurements made with the commercial heat flux sensor are quite noisy. The process is repeated at different temperatures (different heat fluxes) to obtain several points to calibrate the sensor.

The result is shown in Figure 10. The temperature difference as a function of heat flux follows a linear trend. The thermal resistance is obtained from the slope of the regression line. A value of 19 K/W was obtained in good agreement with the simulated estimation. The small difference between simulated and measured $R_{th}$ may be due to irregularities in the assembly.

Once calibrated, to study the concordance between both sensors, a statistical analysis was performed with two different sets of measurements to ensure the reliability of the sensor and of the calibration. First, Pearson’s correlation coefficient was calculated. This method shows the degree of correlation between two variables (how strongly two pairs of variables are associated). This statistical analysis can be applied when the sensors have a linear relationship. However, if this relationship were not linear, the results might be inconclusive or even misleading. The correlation coefficient ranges from −1 to 1. The closer the coefficient value is to zero, the lower the strength of the linear relationship, 1 is the perfect correlation and −1 is the inverted perfect correlation.

Good correlation does not mean good agreement. Despite that these two concepts are used to assess the strength of association between different data sets, they are different from each other. Correlation evaluates the relationship of the trends of data fluctuations. If both variables increase following a linear relationship, regardless of the absolute value of the variables a good correlation factor will be obtained. Consequently, Pearson’s analysis can be considered valid but not sufficient by itself, especially in this case, where the agreement is much more important than the correlation.

Therefore, to assess the validity of the 3D-printed sensor compared to the commercial one, a second statistical method, Bland–Altman analysis, will be used to process the measurements. This statistical method evaluates how the probability that one set of measurements differs from another one by means of a graphical method. The X-axis represents the average and the Y-axis represents the difference in the measurements. The limit of agreement, or the standard deviation, is set to ±1.96, which represents a 95% confidence interval of distributed data.

The statistical analysis was done for two different cases. The first one consists of the measurement of a wide range of heat fluxes in a short period of time. The resulting correlation coefficient between both measurements is 0.9978. Figure 11 shows the measurement of the heat flux of the two sensors, from 500 to 0 W/m${}^{2}$, for 90 min. Figure 12 shows the Bland–Altman analysis. The resultant mean bias was 0.00 with a standard deviation of 11 and −11 W/m${}^{2}$. The commercial sensor data have a higher noise rate compared to the 3D-printed sensor. This is because the commercial sensor is based on an array of series-connected thermopiles that provides a few microvolts, which are amplified by a differential amplifier and sampled with a 10-bit resolution analog-to-digital converter (ADC). The noise of this commercial sensor increases considerably for high values of heat flux, which increases the standard deviation of the Bland–Altman analysis. Scatter can be reduced by smoothing the measurements of this commercial sensor.

The second data set consists of a long-term measurement with the sensors installed on a wall, identical to how the U-value of a building envelope would be assessed. The resultant correlation coefficient between both measurements in this case is 0.995. Figure 13 shows the heat flux measurements for a period of eight days. It can be seen that the fluctuation range of the heat flux is now much smaller, from −10 to 15 W/m${}^{2}$. Figure 14 shows the Bland–Altman analysis. The resultant mean bias was −0.01 with a standard deviation of 0.86 and −0.88 W/m${}^{2}$.

Finally, both the Pearson’s correlation coefficient and the Bland–Altman analysis show a good correlation and agreement between the two sensors. Although the error increases considerably for high heat fluxes due to the ripple of the commercial sensor, when the data are averaged, the mean values of both sensors show a good agreement, reducing the error to almost half.

The limit temperature of the PLA before its deformation is 60 °C and the temperature sensors (MAX30205) operate from 0 °C to +50 °C. Therefore, the heat flux sensor can operate within this temperature range that is suitable for the proposed application (U-value measurement) considering that it is installed indoors. The sensibility obtained from the calibration procedure is $1/(R_{th}\cdot A)$ = 26.8 W/(m${}^{2}$K). The MAX320205 converts the temperature measurements to digital format using a high-resolution, sigma-delta, 16-bit ADC. Therefore, the temperature resolution is $\Delta T_{min}$ = 0.00390625 °C, which corresponds to a heat flux resolution of $\Delta T_{min}/\left(R_{th}\cdot A\right)$ = 0.1 W/(m${}^{2}$K). From the measurements of the heat flux over time, the standard deviation has been obtained with the heat flux sensors installed in a window. Assuming that the heat flux does not change over the measurement time (15 min), the estimated signal-to-noise ratio for the proposed flux sensor is 25.4 dB, compared to 21.8 dB achieved by the commercial sensor. In order to study the repeatability, the measurement was repeated 10 times, each time installing the sensor in the same double-glazed window, achieving a standard deviation of 1.2 W/m${}^{2}$ and a maximum difference of 0.2 W/m${}^{2}$ over the samples.

## 4. Results

The metering node has been installed in two different places to assess the U-value. The first measurement point where the system was installed was a double-glazed insulated glass of the SGG Climalit brand. The second measurement point was a masonry brick wall of a house façade built in the 1990s in Spain.

The U-value measurement node has been installed at each measurement point together with the commercial sensor, very close to each other to ensure that both are exposed to the same heat fluctuations. The commercial sensor provides a redundant measurement of heat flux that can be used to further validate the measurement of the 3D-printed sensor. Sensors were attached to the wall by means of a thermal interface sheet, which reduced the air gaps and improved the heat transfer between the sensor and the wall. Figure 15 shows a photograph of the U-value meter installation in the first scenario. The locations of the outdoor temperature sensor have been carefully selected to avoid direct exposure to sunlight or wind currents. The LoRa gateway has been installed in another room at a distance of about 50 m.

Figure 16 shows the results obtained for a measurement performed over 6 days in the first measurement point, the double-glazed glass. Subplot (a) shows the indoor and outdoor temperatures. Subplot (b) shows the heat flux measurement of the 3D-printed sensor and the commercial sensor. Subplot (c) shows the U-value determined with each of the two heat flux measurements. The average U-value obtained for both sensors is the same, 2.485 W/m${}^{2}$·K. Typical U-values for double-glazed windows range from 1.5 to 3.5 W/m${}^{2}$·K [42,43,44].

When the inside and outside temperatures intersect, the heat flux is reversed, which makes it necessary to determine the U-value by sections or by averaging all the measurements. The larger the temperature difference, the more accurate the resultant U-value. Referring to ISO standards, to obtain an accurate U-value, the temperature difference must be greater than 5 °C. Maintaining a temperature difference greater than 5 °C for a long time can be difficult or directly impossible depending on the season of the year if active heating or cooling systems are not available. In such cases, the U-value must be determined by sections or by averaging the measurements, as proposed in the application case detailed by the company greenTEG [45].

Figure 17 shows the U-value measurements of a masonry brick façade. The system has been installed inside, and the measurements have been taken during the summer season in Spain with temperatures above 30 °C. In this case, as in many other places where U-value evaluations must be carried out, there is no air conditioning system that allows us to control indoor temperature. Therefore, as expected, the direction of the heat flux changes in measurements taken at night or during the day. In this case, the U-value must be assessed by sections during the night periods since the temperature gradient is higher.

Due to the climatic conditions and the lack of active air conditioning systems, the sections with temperature differences greater than 5 °C are very narrow and have very few points. In order to determine a more reliable U-value, sections with temperature differences greater than 4 °C have been evaluated, thus increasing the number of averaged points. Results are presented in Table 5.

Finally, averaging the U-values obtained in each section, we get a U-value of 0.313 W/m${}^{2}$K. The Spanish building technical code stipulates a maximum U-value of 0.41 W/m${}^{2}$K [46,47]. Therefore, the value obtained complies with current regulations.

## 5. Conclusions

The need to improve the thermal insulation of the buildings requires on-site thermal performance measurement and analysis of building materials and structures. A heat flux meter is recommended as standard equipment for performing these measurements. However, commercial heat flux meters are expensive and impractical when multiple measurement points need to be evaluated.

A LoRaWAN-based wireless node for U-value assessment with a low-cost 3D-printed heat flux meter has been presented in this paper. A procedure for the calibration of the proposed heat flux sensor has been presented. Good agreement and correlation have been obtained over different ranges and measurement periods compared to standard commercial heat flux sensors based on thermopiles.

The high sensitivity of LoRa transceivers, as well as the robustness against interference, enables long-range indoor communications by overcoming attenuation introduced by walls and multipath fading. Point-to-point wireless communication via LoRa has been tested in a real environment within unfavorable conditions, obtaining good results. Any of the nodes that are within the range of an internet access point can act as a gateway, collecting all the data from the other nodes and sending it to the host over Wi-Fi using an MQTT protocol. Custom or public MQTT brokers can be used for this purpose. Therefore, the designed platform improves flexibility compared to heat flux meters based on conventional data loggers.

Finally, the evaluation of the thermal transmittance has been carried out in two different scenarios, obtaining good results.

## Figures and Tables

**Figure 1 sensors-22-07259-f001:**
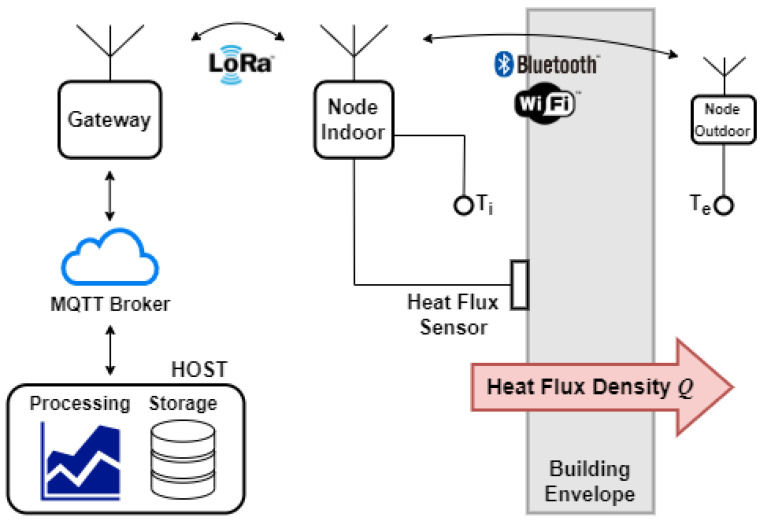
In situ U-value measurement setup for an average method according to ISO9869:2014 and remote data acquisition based on LoRaWAN network and MQTT protocol.

**Figure 2 sensors-22-07259-f002:**
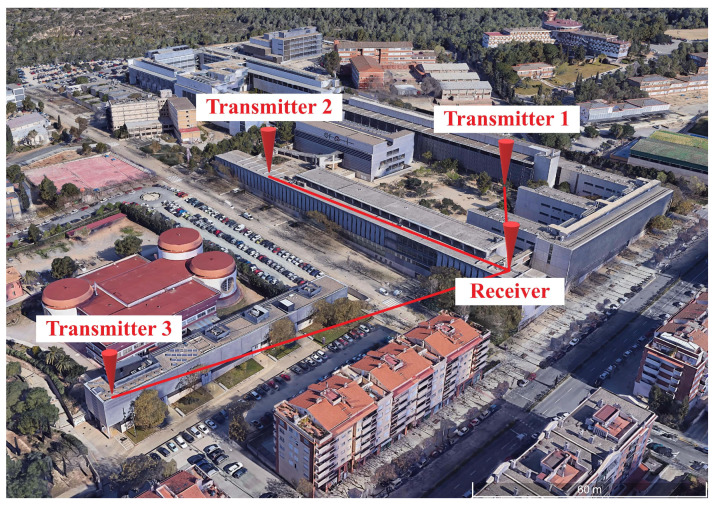
LoRa transceiver deployment.

**Figure 3 sensors-22-07259-f003:**
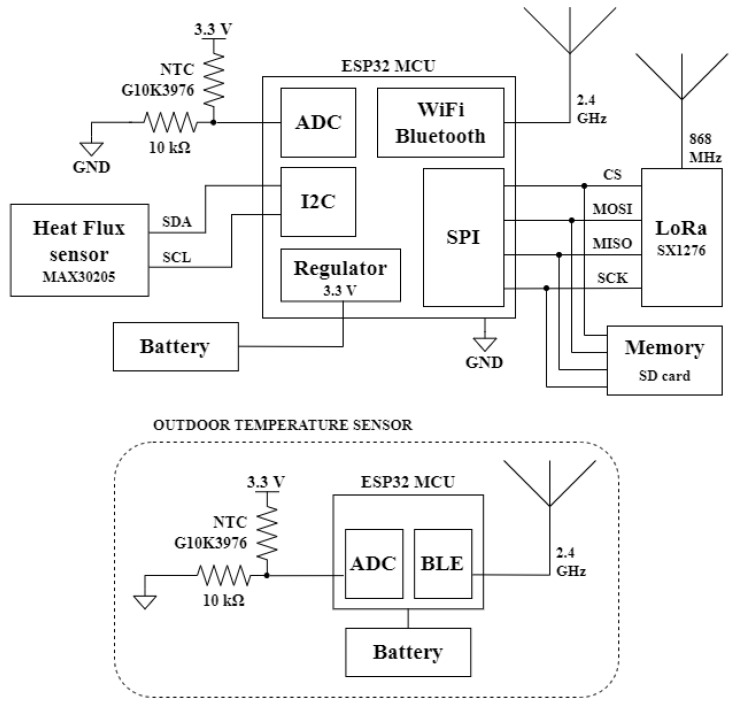
Block diagram of the U-value measurement system.

**Figure 4 sensors-22-07259-f004:**
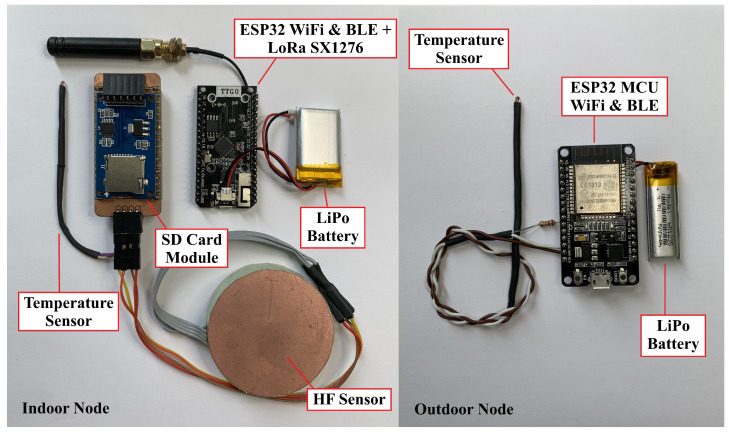
Photograph of the U-value measurement prototype.

**Figure 5 sensors-22-07259-f005:**
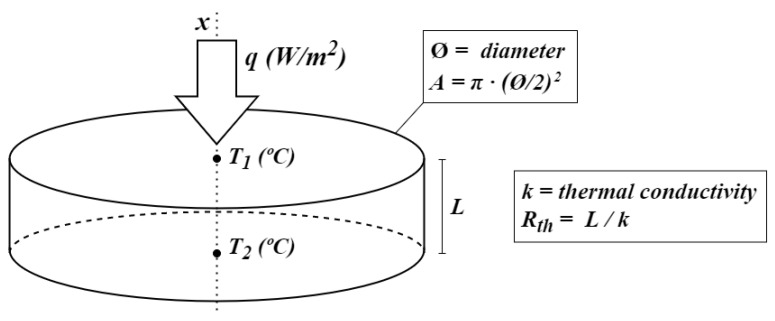
Heat flux sensor diagram.

**Figure 6 sensors-22-07259-f006:**
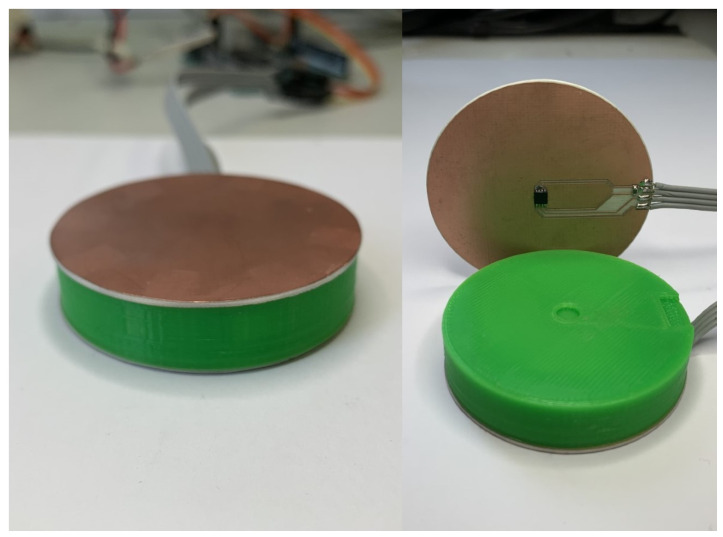
Heat flux sensor prototype.

**Figure 7 sensors-22-07259-f007:**
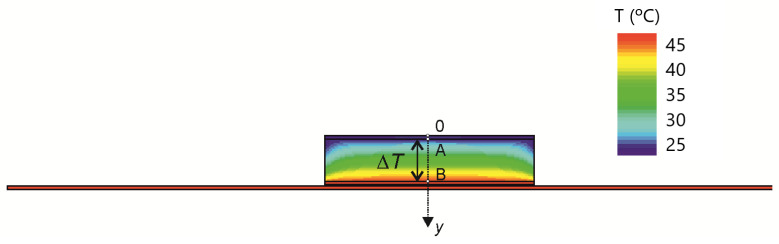
Simulated temperature distribution of a cross-section of the sensor.

**Figure 8 sensors-22-07259-f008:**
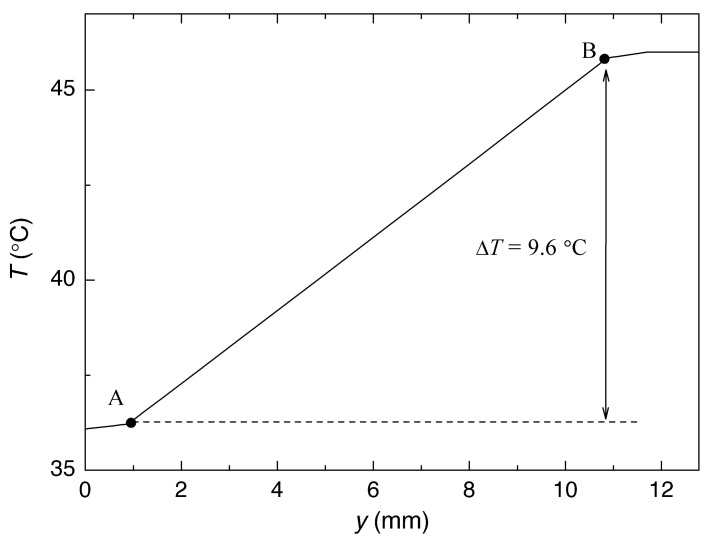
Temperature variation in depth (along the axis of the sensor).

**Figure 9 sensors-22-07259-f009:**
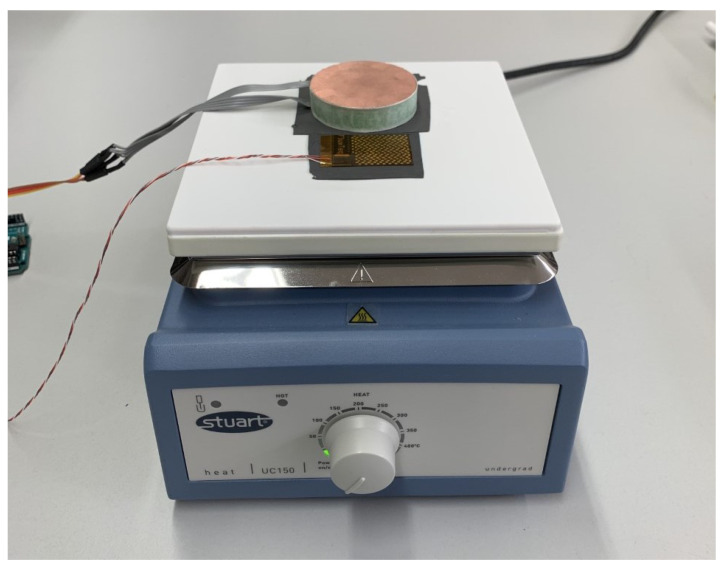
Calibration setup: UC150 heater, Fluxteq PHFS-01 sensor, and 3D-printed HF sensor.

**Figure 10 sensors-22-07259-f010:**
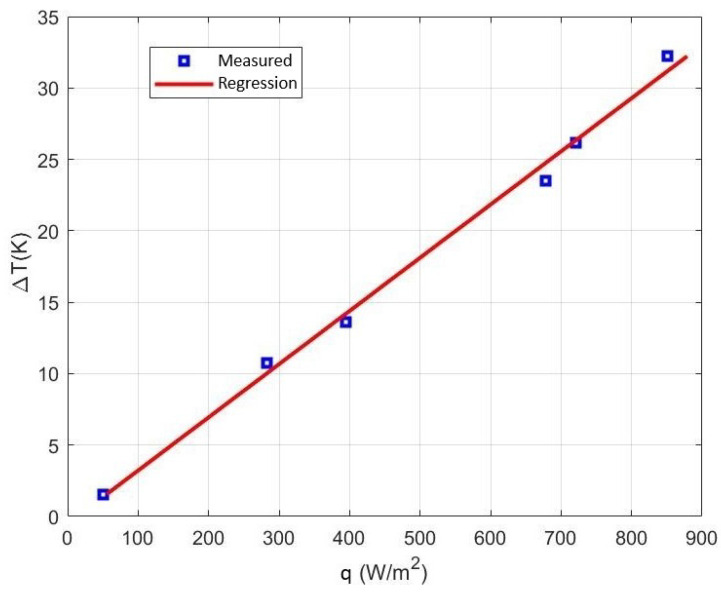
Temperature gradient between both sides of the 3D-printed sensor as a function of the applied heat flux.

**Figure 11 sensors-22-07259-f011:**
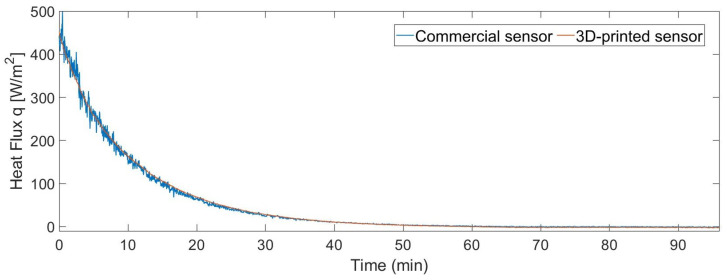
Comparison between the measurements of the commercial sensor and the 3D-printed sensor for a wide range of heat fluxes and a short period.

**Figure 12 sensors-22-07259-f012:**
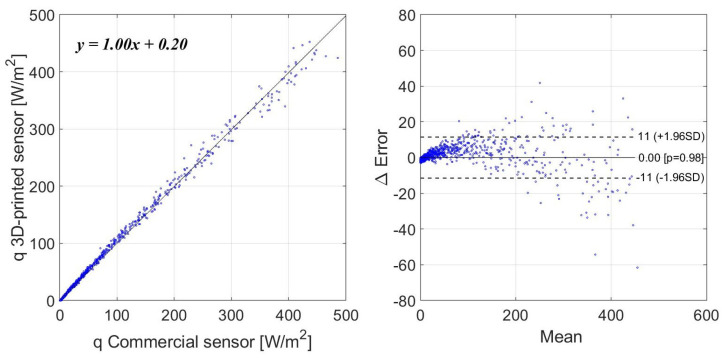
Bland–Altman analysis for short-term measurements and a wide range.

**Figure 13 sensors-22-07259-f013:**
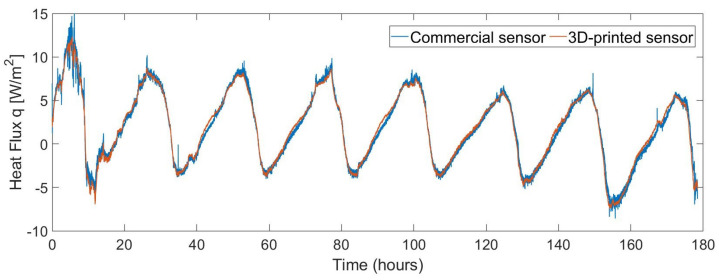
Comparison between the measurements of the commercial sensor and the 3D-printed sensor installed on a wall for eight days.

**Figure 14 sensors-22-07259-f014:**
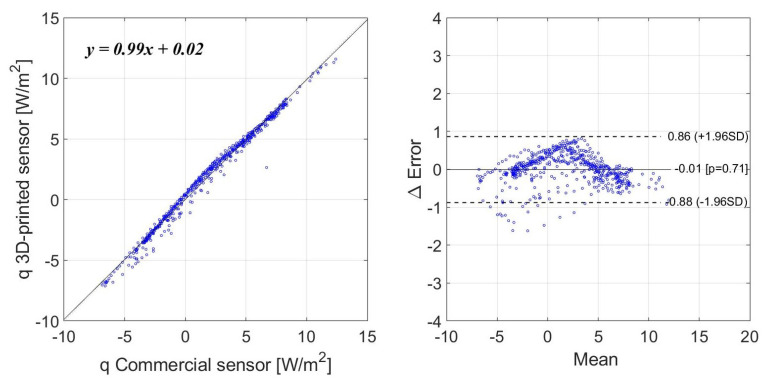
Bland–Altman analysis for long-term measurements and a narrow range.

**Figure 15 sensors-22-07259-f015:**
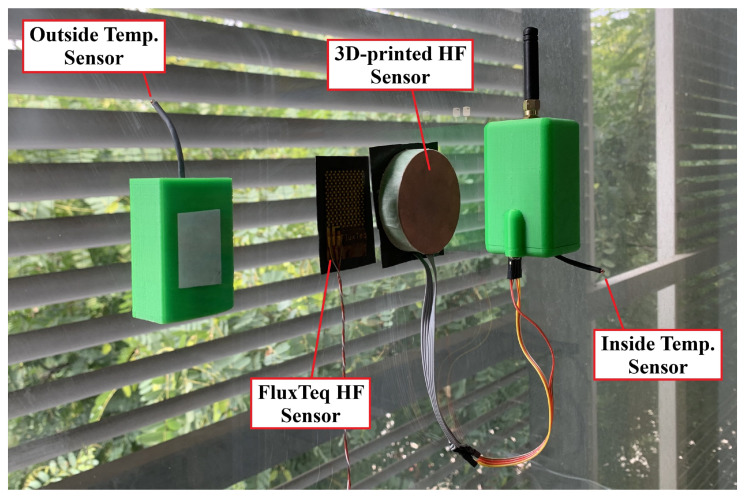
Photograph of the measurement nodes installed in a large double-glazed insulating glass. The main node has been installed inside and the outdoor temperature sensor has been attached outside, avoiding direct exposure to sunlight.

**Figure 16 sensors-22-07259-f016:**
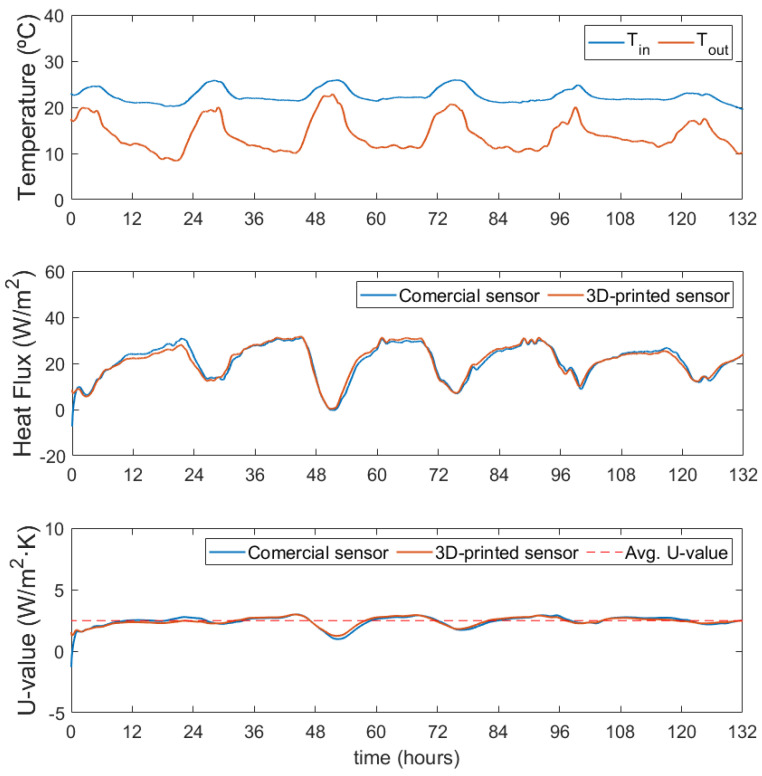
Evaluation of the U-value of a double-glazed insulated glass window for six days.

**Figure 17 sensors-22-07259-f017:**
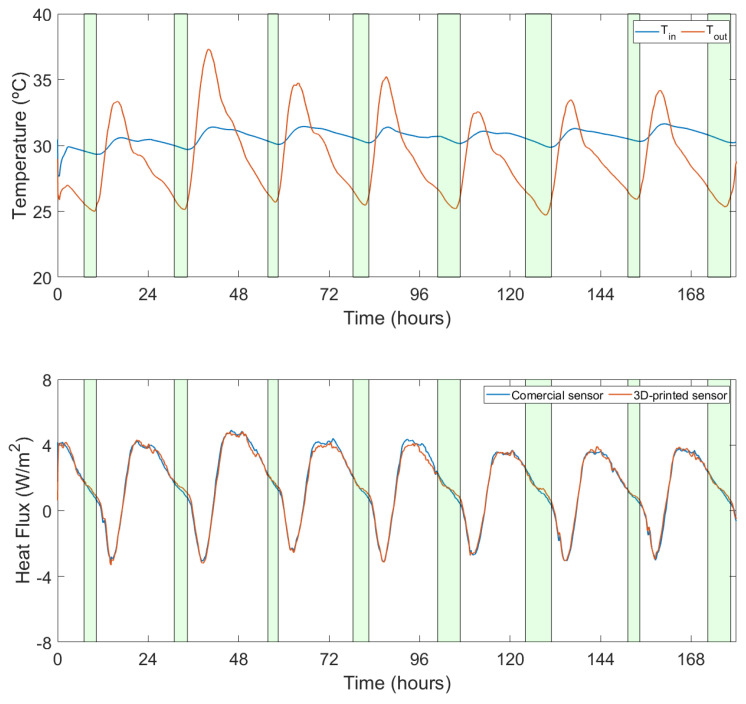
Evaluation of the U-value of an exterior masonry brick wall for five days.

**Table 1 sensors-22-07259-t001:** LoRa communications performance in a real environment.

Receiver	Distance	RSSI	SNR	Packet Loss
Transmitter 1	100 m	−124 dBm	−12.7 dB	4.9%
Transmitter 2	130 m	−122.7 dBm	−14.9 dB	6.8%
Transmitter 3	150 m	−123.5 dBm	−0.77 dB	0 %

**Table 2 sensors-22-07259-t002:** Current consumption of the wireless modules and battery lifetime.

Value	Indoor module at 13 dBm LoRa Tx Power	Indoor module at 20 dBm LoRa Tx Power	Outdoor (ESP32 Devkt 1)	Outdoor module (FireBeetle ESP32)
Active mode current (mA)	79	170	100	100
Deep-sleep mode current (mA)	0.2	0.2	5	0.15
Average current (mA)	1.51	3.0	6.58	1.8
Battery capacity (mAh)	2000	2000	2000	2000
Battery lifetime (days)	27	27.5	12	45.9

**Table 3 sensors-22-07259-t003:** Comparison of thermal conductivity of thermoplastic used in 3D printers.

Material	Thermal Conductivity, k (W/m·K)	Thermal Expansion (μm/m·K)	Glass Transition Temperature (°C)
PLA	0.13	68	60
ASA	0.17	95	100
ABS	0.25	90	105
PETG	0.29	68	85
PEEK	0.25	60	143

**Table 4 sensors-22-07259-t004:** Simulation settle.

Material	Thermal Conductivity, k (W/m·K)
Copper	385
PCB	0.71
PLA	0.13
Aluminium	238
Air	0.025

**Table 5 sensors-22-07259-t005:** Average U-value by section.

Sections	Date	Time	U (W/m${}^{2}$K)	Variation
Night 1	13.07	3:17–6:31	0.3170	1.257%
Night 2	14.07	3:10–6:43	0.3202	2.245%
Night 3	15.07	4:02–6:45	0.4209 *	>5.00%
Night 4	16.07	2:36–6:49	0.3152	0.695%
Night 5	17.07	1:24–7:33	0.2998	−4.425%
Night 6	18.07	0:19–7:13	0.3153	0.694%
Night 7	19.07	3:30–6:42	0.2160 *	>5.00%
Night 8	20.07	0:43–6:48	0.3108	0.737%
Average	-	-	0.3131	-

* In compliance with ISO, deviations higher than 5% are not considered for the analysis.

## Data Availability

The data presented in this study are available from the corresponding author upon request.

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
