# Peer review of "Long-Range Wireless System for U-Value Assessment Using a Low-Cost Heat Flux Sensor"

_sensors, 2022, doi:10.3390/s22197259_

Round 1

Reviewer 1 Report

A LoRaWAN-based wireless node for U-value assessment with a low-cost 3D-printed heat flux meter has been presented in this paper.A procedure for the calibration of the proposed heat flux sensor has been presented. Good agreement and correlation have been obtained over different ranges and measurement periods compared to standard commercial heat flux sensors based on thermopiles. These efforts have been very productive, It is recommended that the paper be accepted with some minor revisions.

1.P2 Line 57 and Line 59,Is TFM wrong. TBM?

2.Letter abbreviations and variable letters should require a symbol table. The first time an abbreviation appears it should give a specific meaning.

3. During the test, the distance between the receiver and the test point should be given.

Reviewer 2 Report

The manuscript attempts to present a case about a long-range wireless system for U-value assessment in building envelopes using a 3D-printed Heat Flux Sensor. The paper is well-written and has the style and language demanded for a potential publication. The list of references should definitely be expanded, since they are few in quantity even if it is a research paper.

My points are analytically listed below

Points for consideration:

Point 1: My biggest argument is derived from the title. Judging by paragraph 3.1 the sensor itself is not 3D printed. Its casing is. This can be largely misleading for the readers since the title implies that the sensor is printed (perhaps it could have been with a metal 3D printer). If you cannot change the title (this would be the right thing to do), at least make a clarification in the abstract or the introduction.

Point 2: Even though the words “3D printed” are included in the title, there is not a single paragraph about 3D printing or pictures from the actual 3D printing of the sensor’s casing. I strongly believe that the authors should write one or two paragraphs about desktop FDM/FFF 3D printers, like the one probably used in the fabrication of the PLA casing. Take a look at these relevant papers and include them in your references that also need to be expanded.

·         10.1016/j.matpr.2020.08.627

·         10.5923/j.mechanics.20211001.02

·         10.3390/ma12121970

·         10.3390/technologies9040091

Point 3: In section 3.1 you mention that the material used is PLA. For the service temperatures mentioned for the sensor (15-450C) it is ok. However, judging by figure 15, PLA for external environment use should be avoided. Especially with the weather of a mediterranean country like Spain, the PLA casing will warp and deform under the direct sunlight. Have the authors considered using materials like PETG, ABS or even better ASA that can withstand external weather conditions? This would strengthen the value of the paper.

Wishing the best for your paper,

The reviewer.

Reviewer 3 Report

For large-scale building envelope heat flow measurements, the authors suggest a low-cost, wireless, long-range communication-based heat flow sensor. A commercial sensor was used for calibration, and field testing was done to compare the accuracy of U-value measurements with the commercial sensor. It is advised that the writers explain or discuss the following issues in the text to better grasp state of the art.

1. In addition to long-range communication, there are other considerations for the wireless communication system proposed in the title and abstract, such as low power consumption and networking mode for deploying multi-point measurement systems applied to a wide range of building maintenance structures.

2, whether this portion has the relevant experimental or theoretical analysis, sensitivity, heat flow resolution, heat flow measurement range, noise equivalent heat flow, operating temperature range, and repeatability are all essential concerns for the heat flux sensor.

3, Does the ESP32 processor ADC's precision match the standards, or does it require additional improvement?

Round 2

Reviewer 3 Report

I have no additional comments and agree that the article is accepted and recommended for publication